

# Variation in fine-scale water table depth drives abundance of a unique semi-terrestrial crayfish species

Molly C. Carlson[1,2], Dusty A. Swedberg[2], Elizabeth A. Miernicki[2] and Christopher A. Taylor[2]

[1] Department of Natural Resources and Environmental Sciences, College of Agricultural, Consumer and Environmental Sciences, University of Illinois at Urbana-Champaign, Champaign, IL, United States of America

[2] Illinois Natural History Survey, Prairie Research Institute, University of Illinois at Urbana-Champaign, Champaign, IL, United States of America

Corresponding author
Molly C. Carlson, mcc12@illinois.edu

## ABSTRACT

With anthropogenic changes altering the environment and the subsequent decline of natural habitats, it can be challenging to predict essential habitats for elusive and difficult to study taxa. Primary burrowing crayfish are one such group due to the complexity in sampling their semi-terrestrial, subterranean habitat. Sampling burrows usually requires a labor-intensive, time-consuming excavation or trapping process. However, limited information on burrowing crayfish suggests that fine-scale habitat variation may drive burrowing crayfish habitat choice. This project aimed to evaluate the fine-scale habitat characteristics that influence burrowing crayfish presence and abundance at a large, restored-remnant grassland preserve in north-central Illinois. We documented burrow abundance and quadrat-specific habitat variables such as root biomass, canopy cover, apparent seasonal high-water table (water table) depth and dominant vegetation at sites with and without burrowing crayfish populations. Data was recorded at every quadrat and analyzed using generalized linear mixed models. A total of 21 models were created to determine what habitat variables affected burrow presence and abundance. We found that the water table depth was a significant driver of burrow presence and abundance. Root biomass and vegetation cover were not significant drivers, although they did show up in the final models, explaining the data. These findings demonstrate empirical support for previous observations from other burrowing crayfish research and demonstrate the influence of fine-scale habitat when modeling elusive taxa requirements.

# INTRODUCTION

Effective conservation and management of a species depends on a comprehensive understanding of its distribution, and habitat associations (*Whittaker et al., 2005*; *Alzate & Onstein, 2022*). However, this understanding is currently lacking for an array of taxa globally (*Whittaker et al., 2005*; *Lomolino et al., 2010*). These knowledge gaps pose a significant challenge to conservation biology, limiting our ability to predict a species' response to

environmental changes (*Whittaker, Willis & Field, 2001*; *Ackerly et al., 2010*). Studying all biogeographical aspects for a single species is complex, and often limited on a temporal and spatial scale (*Gillson & Willis, 2004*; *Lomolino et al., 2010*). As a result, research efforts are often refined, focusing on a set of limiting factors associated with the target species' distribution and habitat associations (*Lomolino et al., 2010*). While this approach does not fill all the gaps in our knowledge, it can yield useful new insights into species' needs. The demand for such insights becomes more pressing in a world where climate change, urbanization and development directly threaten unique habitats (*Ackerly et al., 2010*).

Anthropogenic alterations threaten habitats like prairies and wetlands, which support a significant amount of global biodiversity, rivaling that of tropical rainforests (*Petermann & Buzhdygan, 2021*). Primary burrowing crayfish are one group of elusive species that are frequently linked to prairie and wetland habitats (*Taylor et al., 2007*; *Welch et al., 2008*; *Reynolds et al., 2013*). However, monitoring these crayfish can be difficult, in part due to their fossorial behavior. While all crayfish can burrow, they are classified into three groups based on their burrowing tendencies: primary, secondary, and tertiary burrowers (*Hobbs, 1942*). Primary burrowing crayfish species, hereafter referred to as burrowing crayfish, spend most of their lives in burrows consisting of a tunnel and chamber system providing protection, shelter, and access to groundwater (*Hobbs, 1942*; *Hobbs Jr, 1981*). These burrows can be far from permanent water bodies, often in prairie and wetland habitats (*Hobbs Jr, 1981*). Since burrowing crayfish are fossorial, with burrow openings usually occurring in vegetated areas, their detection and subsequent sampling can be difficult. Burrows can reach more than 2 m in depth and while several sampling methods have been tested, such as baiting and trapping, hand excavation has proven to be the most effective method (*Norrocky, 1984*; *Welch & Eversole, 2006a*; *Ridge et al., 2008*; *Loughman, Foltz & Welsh, 2013*). However, this method requires excavating the burrows with small trowels, which can be labor intensive, and result in limited success (*Ridge et al., 2008*). As such there is limited information on burrowing crayfish.

The information available demonstrates burrowing crayfish's historic use of riverbanks, prairies, and wetlands (*Hobbs, 1942*; *Hobbs Jr, 1981*). However, today they are also frequently found in roadside ditches, floodplains, and manicured lawns which serve as wetland microhabitats within otherwise unsuitable landscapes (*Rhoden, Taylor & Peterman, 2016*; *Bloomer, Distefano & Taylor, 2021*; *Bearden et al., 2022*). Burrowing crayfish are rarely uniformly distributed within these habitats. Previous studies have examined these distribution and habitat variables, such as the number of active burrows, site level diversity, distance to flowing water, and other surface-level habitat characteristics (*Taylor & Anton, 1998*; *Helms et al., 2013*; *Rhoden, Taylor & Peterman, 2016*). While these studies document a wide range of variables within and between landscapes, they do not document variation within a landscape at a very fine scale that may drive burrowing crayfish distribution. Likewise, previous work has purported that certain habitat variables limit primary burrowing crayfish, but these have not been well researched. For instance, it has long been hypothesized that primary burrowing crayfish rely on root systems for burrow stability and diet, however no study, to date has investigated this relationship within a landscape. The correlation between water table depth and burrow location has also been

![PeerJ]

suggested, but few studies have tested this relationship and found mixed results (*Hobbs, 1942*; *Hobbs Jr, 1981*; *Welch & Eversole, 2006b*; *Bearden et al., 2022*). *Bearden et al. (2022)* is one of few studies to document the relationship between active burrowing population and the water table within a landscape using monitoring wells. While this method was effective, it was used across floodplain landscapes and may not be suitable for all conservation areas or prairie-dominant landscapes.

The fundamental gaps in our understanding of burrowing crayfish biogeography must be filled to determine their overarching conservation needs (*Moore, Distefano & Larson, 2013*). It is imperative to determine current habitat associations for burrowing crayfish to better understand future landscape connections and distributions, as access to vital habitat continues to fluctuate at both local and global scales. To understand such ecological dynamics, we must consider the spatial scale at the target species level (*Wiens, 1989*). While habitat variables will influence burrowing crayfish at several scales, the fine-scale lens provides a focused look into these dynamics on a burrow specific level. Through a localized study into burrowing crayfish's habitat associations, we can determine what limiting factors drive their distributions and gain broader implications for conservation efforts.

We aim to evaluate the fine-scale habitat characteristics that may affect the Great Plains Mudbug (*Lacunicambarus nebrascensis*) (*Girard, 1852*), a burrowing crayfish species, within a north-central Illinois prairie nature preserve, The Nature Conservancies Nachusa Grasslands (Nachusa) in Ogle and Lee counties. By collecting select habitat variables throughout Nachusa, we aim to determine what drives burrowing crayfish distribution at a fine scale. We expect shallow water table depth, high root biomass, and low canopy cover to predict burrow presence and abundance.

## MATERIALS & METHODS
### Study sites
Nachusa was chosen for its unique ∼1619 ha community of restored and remnant habitat consisting of prairies, wetlands, savannas, and forests. Tilling and tile drainage are frequently used in the surrounding landscape to allow for the wet soil to accommodate the cultivation of agricultural products such as corn, beans, and hay (*Elmer & Zwicker, 2005*; *Elmer & Boggess, 2008*). Some areas within Nachusa were formally tiled and harvested annually, but much of the preserve is remnant native land dominated by prairies and wetlands (*Bach & Kleiman, 2021*). The area is characterized as a rolling glacial till plain commonly found within the northwestern Illinois region (*Elmer & Zwicker, 2005*).

We selected six sites within Nachusa to assess burrowing crayfish populations. These sites were selected based on their habitat potential and include remnant and restored habitats (Fig. 1 & Table 1). The restored locations vary in maturity, from recently established to well established secondary-succession communities. Native planting and restoration started in 1994 for site 1 and between 1991 and 1992 for site 2 (Table 1). Sites 4 and 5 are more recent additions to Nachusa with native planting and restoration starting in 2021 and 2010 respectively (Table 1). Sites 3 and 6 were remnant native habitats when acquired by Nachusa (Table 1). Prior to Nachusa's ownership the restored sites were tile drained and harvested.

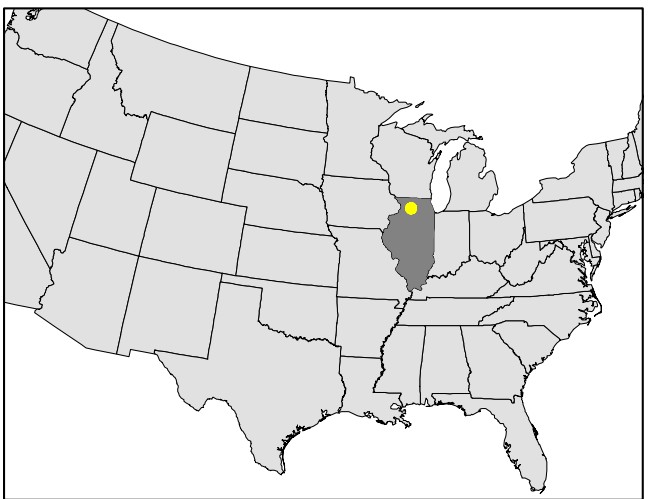

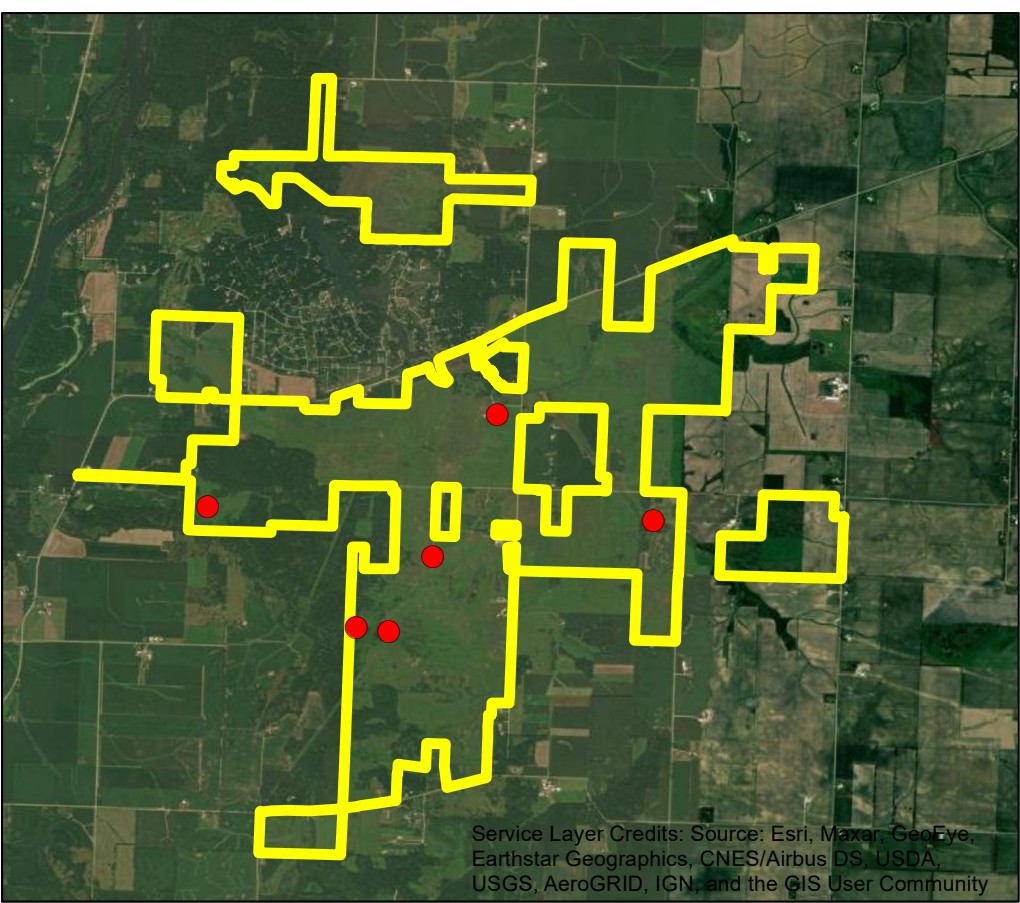

**Figure 1 The six sampling sites at the Nachusa Grasslands.** Aerial map of the six sites that were sampled at the Nachusa Grasslands in Ogle and Lee counties, Illinois in relation to the United States of America. Portions of this document include intellectual property of Esri and its licensors and are used under license. Copyright ©2024. Esri and its licensors. All rights reserved.

**Table 1 Description of each sample site and burrow abundance.** The six sites sampled within Nachusa, the description of their restoration status as of the time of sampling, the burrow abundance recorded per site and the active burrow descriptions.

| Sample Site | Restoration Status | Active Burrow Abundance | Average Number of Burrow Openings | Average Diameter of Burrow Openings |
|---|---|---|---|---|
| Site 3 | Remnant prairie/wetland | 89 burrows | 3 openings | 24.06 mm |
| Site 6 | Remnant prairie/wetland | 21 burrows | 1 opening | 19.40 mm |
| Site 1 | Restoration started in the early 1990's | 17 burrows | 2 openings | 31.64 mm |
| Site 2 | Restoration started in the early 1990's | 12 burrows | 1 opening | 29.44 mm |
| Site 4 | Restoration started in 2020. | 7 burrows | 2 openings | 25.88 mm |
| Site 5 | Restoration started in 2010. | 0 burrows | 0 openings | 0 mm |

An initial species assessment was conducted in early May 2022 at each of the six sites to determine what species were present. We excavated active burrows if present and used timed kick seining in streams flowing near the sites. The only burrowing species collected within Nachusa was *L. nebrascensis*, these specimens were identified in the field and confirmed in the lab using the species description (*Glon et al., 2022*). Of the six sites selected, five had active burrowing crayfish populations (Fig. 1 & Table 1).

## Field collection

We conducted field sampling during the spring and summer months of 2022 (May–July), all field work was approved by the Illinois Chapter of the Nature Conservancy and the Illinois Nature Preserves Commission. We used a standard transected design to collect habitat-specific variables at each site (*Rhoden, Taylor & Peterman, 2016*; *Adams, Hereford & Hyseni, 2021*). We laid a 100 m baseline transect at a randomly selected starting point and ran along the lowest elevation in each site. Six 100 m sampling transects were laid perpendicular to the baseline. We randomized the direction (left or right) of the first sampling transect with subsequent transects alternating directions unless we encountered accessibility issues (Fig. 2). To ensure sampling transect independence, we set a minimum distance of 10 m between each sampling transect. However, we determined the exact distance by conducting a burrow search along the baseline within an additional 5 m, laying the subsequent sampling transect at the closest burrow. If we encountered no burrows, we laid the next sampling transect 15 m (10 m minimum distance with an additional 5 m for burrow search) from the previous sampling transect (Fig. 2).

A total of six quadrats, 20 m apart, were placed along each 100 m sampling transect. We used a 1 m$^2$ polyvinyl chloride (PVC) quadrat to assess habitat variables. To account for the potentially low number of burrows found within a 1 m$^2$ quadrat, a larger 9 m$^2$ quadrat area was assessed for burrow abundance. We did this by flipping the 1 m$^2$ quadrat around the perimeter of the initial quadrat area, summing the number of burrows within each section.

We recorded active burrows found within each 9 m$^2$ quadrat area and documented their activity level. Active burrow presences and abundance (number of burrows) was recorded at every quadrat. We defined burrows as active if fresh moist soil was present around the

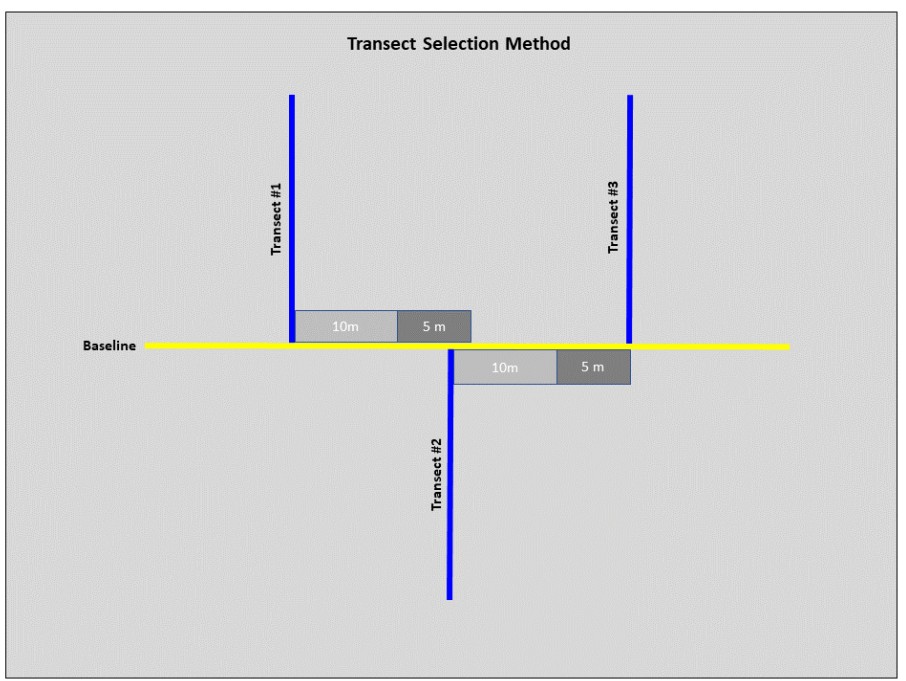

**Figure 2** **Diagram of the selection method for transects at each site.** From transect #1, 10 m were measured (light grey), and a 5 m burrow search was conducted along the baseline (dark grey). If a burrow was before the 5 m burrow search was concluded, then this burrow marked the start of transect #2. From transect #2, 10 m were measured, and another 5 m burrow search was conducted. If no burrows were detected by the end of the 5 m search, this marked the start of transect #3. These steps were repeated for subsequent transects.

opening of the burrow, the entrance of the burrow was clear, or had a fresh plug or chimney around the opening (*Helms et al., 2013*).

## Habitat variables

We recorded fine-scale habitat variables at each 1 m$^2$ quadrat. These include vegetation ground cover, dominant vegetation type by family, dominant plant functional group, canopy cover, root biomass, apparent seasonal high-water table (water table) depth and soil texture at the water table (Table 2).

We used a modified version of the Daubenmire method to measure the percent of space within the quadrat occupied by vegetation (*Daubenmire, 1966*; *Stohlgren et al., 1997*). This was taken by estimating the percentage of space each plant occupied within the total area of the quadrat. The dominant vegetation by families and plant functional groups were identified within each quadrat using field identification guides (*Peterson & McKenny, 1968*; *Struwe, 2009*), and the mobile app Seek by *iNaturalist (2022)*. Similarly, dominant plant functional groups were categorized as annual/biannual forbs, spring, ephemeral spring and summer/fall forbs, warm season graminoids ('C' grasses), cool season graminoids ('C3' grasses), legumes and woody shrubs (*Kindscher & Wells, 1995*). We measured canopy cover

**Table 2 Habitat variables measured and a description of each measurement.** The habitat variables we measured at every quadrat and a description of the method used to measure each one.

| Habitat Variable | Description |
|---|---|
| Percent vegetation cover | % space within each quadrat occupied by vegetation was measured using the Daubenmire method (*Daubenmire, 1966*; *Stohlgren et al., 1997*). |
| Percent dominant vegetation by family | Dominant vegetation within every quadrat was identified down to family level. |
| Dominant plant functional group | Groups of plants that share like traits and ecosystem functions were identified within every quadrat (*Kindscher & Wells, 1995*). |
| Canopy cover | % tree canopy cover was measured at every quadrat by counting the space within the spherical densiometer (*Cook et al., 1995*). |
| Root biomass | Root cores were taken at every quadrat using a root auger. Roots were weighed after being washed and dried (*Böhm, 1979*). |
| Apparent seasonal high-water (water table) depth and soil texture | Redoximorphic depletions and soil texture were inspected at every quadrat. The presence of common ($\geq 2\%$) redoximorphic depletions signifies the water table depth (*United States Department of Agriculture, 2018*; *Soil Survey Staff, 2022a*; *Soil Survey Staff, 2022b*). |

by holding a spherical densiometer level at elbow height to ensure we could view the grid, then counted the number of dots within the grid that canopy occupied (*Cook et al., 1995*).

We measured root biomass by taking a 30 cm depth × five cm root diameter core from the top right corner of each quadrat. We wrapped each root core in aluminum foil to keep the cores' structure and stored them in a temperature-controlled room (15–20 °C) until root washing and biomass measurements were taken. We used a pressure washing nozzle to wash cores in sections from bottom to top of the core using a one mm sieve to collect roots (*Böhm, 1979*). We placed the washed roots into separate pre-weighed standard, 22 cm × 11 cm, letter envelopes. We dried roots in the envelopes in an oven at 55 °C for 72 h. After drying, we weighed each envelope, for the final root biomass (g).

We assessed the water table depth in each quadrat by describing a soil core with a JMC soil probe consisting of a 1.9 cm bore (Clements Associates Inc., Newton, IA, USA). Water table depth was determined by using the depth to common grey redoximorphic features (*Illinois Soil Classifiers Association, 2006*). The type of redoximorphic features used to determine water table depth are called depletions, which are formed when iron and/or manganese undergo oxidation and reduction processes. When iron and manganese minerals are reduced, localized zones of "decreased" pigmentation develop and appear grayer, lighter, or less red than the adjacent soil matrix (*Schoeneberger, Wysocki & Benham, 2012*). Redoximorphic features persist in soils during both wet and dry periods (*United States Department of Agriculture, Natural Resources Conservation Service, 2018*). The "common" class for redoximorphic features refers to 2–20 percent of soil surface area covered (*Schoeneberger, Wysocki & Benham, 2012*). This criterion is used in Illinois to determine where the water table is located on average spatially (*Illinois Soil Classifiers*

*Association, 2006*). We recorded the depth to the water table depth, if detected, for each quadrat. If the water table depth was not detected due to the soil probe's range restrictions (≤102 cm) or presence of a limiting layer like bedrock, the lowest depth accessible was recorded.

Soils low in iron or manganese minerals, such as organic soils (*i.e.*, histosols), make water table determinations difficult due to the lack of redoximorphic features (*Soil Survey Staff, 2022a*). Organic matter tends to accumulate in soils under anaerobic conditions, creating thick organic surface horizons, like peat or muck, or dark organic-rich surface layers as a result (*United States Department of Agriculture, Natural Resources Conservation Service, 2018*). Some soil cores taken in Nachusa were from a fen wetland with thick organic surface layers and no identifiable redoximorphic depletions. To determine the water table depth at the sampling points, soil texture and depth were recorded. The texture of these samples was determined by using the texture-by-feel and percentage of visible fibers methods (*Thien, 1979*; *Soil Survey Staff, 2022a*). The Field Indicators of Hydric Soils in the United States (*United States Department of Agriculture, Natural Resources Conservation Service, 2018*) was used as a guide to determine water table depth in the soils that formed under conditions of saturation, flooding, or ponding long enough throughout the growing season to develop anaerobic conditions in the upper part (*Federal Register, 1994*). The organic soil hydric indicators in the guide require Aquic conditions, which refers to soils that currently undergo continuous or periodic saturation and reduction for ≥30 consecutive days in normal years (*Soil Survey Staff, 2022a*).

**Statistical analysis**

We used generalized linear mixed models (GLMMs) in R version 4.2.2 (*R Core Team, 2022*) to model the relationship between the habitat variables and the response variables of burrow presence and abundance. By using GLMMs we could account for the random and fixed effects encountered during field collection. We centered and scaled our variables pre-analysis using a z-score transformation. We did not include any variables with a Spearman correlation coefficient of ≤ 0.60 in the candidate models. We used a binomial distribution to model active burrow presence and absence to account for the finite nature of this variable using the *R* package *glmmADMB* (*Fournier et al., 2012*). We used a zero-inflated Poisson distribution to model burrow abundance due to the infinite nature of this variable using the *R* package *glmmTMB* (*Brooks et al., 2017*). We accounted for site-specific effects in our models by nesting transects within a site as the random effect. We fit the null and global models with the selected predictor variables. We checked for overdispersion in our models using the parameter c-hat, variance from fixed factors using marginal $r^2$ and variance from both fixed and random factors using conditional $r^2$. Candidate models were created and evaluated using Akaike's information criterion (AICc) corrected for small sample size (*Akaike, 1974*). We assessed the relative support for every model and model-averaged parameters for top models using the *R* package *MuMIn* (*Bartoń, 2022*) with ΔAICc values <2.5 units and having majority weight (*Burnham & Anderson, 2002*).

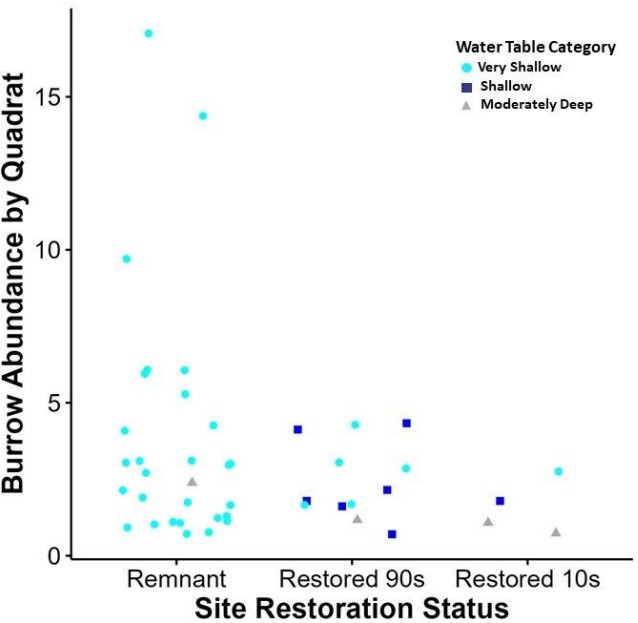

**Figure 3** **Relationship between site restoration status and burrow abundance by quadrate.** Quadrats with positive burrow abundance in relation to the restoration status of the site and the water table depth measurement taken at the quadrat. Water table depth is displayed in categories based on wetland delineation metrics (*United States Department of Agriculture, 2018*). This graph shows some correlation with very shallow water table access (0–25.4 cm in depth), when compared to shallow (25.4–50.8 cm) and moderately deep (50.8–76.2 cm) water table depth. In addition, increased burrow abundance was recorded at sites of remnant habitat than those of secondary succession.

# RESULTS

We documented a total of 146 burrows within the six sites at Nachusa. Of the six sites, the remnant prairie-wetland sites, sites 3 and 6, had the highest abundance (Table 1; Fig. 3). The sites restored in the 1990's, sites 1 and 2, had a moderate burrow abundance (Table 1; Fig. 3). Sites restored between 2010-2020 had the lowest abundance, with site 5 having no burrow abundance (Table 1; Fig. 3). Active burrows had an average of 1–3 openings per burrow across sites (Table 1). Over 70% of the burrows detected throughout Nachusa occurred within the 1st and 2nd quadrats of the transect lines. The top three dominant plant families encountered within the sites were daisies (*Asteraceae*), grasses (*Poaceae*) and sedges (*Cyperaceae*) respectively. The most common plant functional groups encountered were summer/fall forbs.

Of the 21 models created, the top models were Models 1 and 2 for the binomial analysis and Models 5 and 6 in the Poisson analysis ($\Delta$AICc < 2.5) (Table 3). The habitat associations for burrowing crayfish presence and abundance were similar.

The habitat variable with the most influence, indicated by Models 1 and 6, was the water table depth (wt; Tables 3 and 4). The water table depth was negatively associated with burrow presence and abundance (Table 4). Root biomass and vegetation ground cover were positively associated with burrow presence, though not significantly (Table 4). Model

**Table 3  Generalized linear mixed effects model (GLMM) results.** Akaike's Information Criterion adjusted for small sample size (AICc), Akaike weights (wi), and log-likelihood (LL) for the top habitat models (ΔAICc < 2.5) from a suite of variables modeled with a generalized linear mixed-model analysis for burrowing crayfish presence (binomial models) and abundance (Poisson models). Variables included in the table are, water table depth (wt), root biomass (biomass), and vegetation ground cover (vegcov).

| Model Number | Model Variables | AICc | ΔAICc | LL | Wi |
|---|---|---|---|---|---|
| | **Binomial GLMM** | | | | |
| Model #1 | wt | 160.1 | 0.00 | −77.01 | 0.668* |
| Model #2 | biomass +vegcov +wt | 162.6 | 2.49 | −76.16 | 0.192* |
| Model #3 | null | 190.1 | 30.0 | −93.03 | 0.000 |
| Model #4 | global | 172.6 | 12.5 | −71.13 | 0.001 |
| | **Poisson GLMM** | | | | |
| Model #5 | restor+wt | 395.7 | 0.00 | −191.6 | 0.531* |
| Model #6 | wt | 396.5 | 0.79 | −194.1 | 0.358* |
| Model #7 | null | 421.3 | 24.8 | −207.6 | 0.000 |
| Model #8 | global | 408.2 | 13.2 | −163.8 | 0.001 |

**Notes.**
The * indicates the best predictive model from the suite.

**Table 4  Binomial and poisson model averaged parameters.** Model-averaged parameter estimates for the top models for burrow presence and abundance within Nachusa. Models selected from a suite of variables modeled with a generalized linear mixed-model analysis for burrowing crayfish presence and abundance.

| Model Variables | Model-averaged estimate (SE) | 95% CL | P(>\|z\|) |
|---|---|---|---|
| **Binomial Model** | | | |
| Water Table Depth | −1.658(0.380) | −2.337, −0.837 | 1.43e−05* |
| Root Biomass | 0.241(0.231) | −0.211, 0.694 | 0.299 |
| Vegetation Cover | 0.143(0.228) | −0.304, 0.589 | 0.533 |
| Intercept | −1.912(0.532) | −2.925, −0.825 | 0.000356* |
| **Poisson Model** | | | |
| 2010's Restoration Sites | −0.236(0.643) | −1.497, 1.025 | 0.716 |
| 1990's Restoration Sites | 0.6844(0.3805) | −0.061, 1.430 | 0.074 |
| Water Table Depth | −1.288(0.311) | −1.884, −0.787 | 3.48e−05* |
| Intercept | −0.406(0.391) | −1.160, 0.308 | 0.302 |

**Notes.**
The * indicates significant predictor variables.

2 was a top model for the binomial analysis, which included root biomass, vegetation ground cover, and water table depth. However, the only variable of significance for Model 2 was the water table depth (Table 4). Model 5 was a top model for the Poisson analysis, which included restoration status of the site and water table depth. Again, water table depth was the only significant variable in this model (Table 4). Restoration status was positively associated with burrow abundance but not significantly (Table 4).

## DISCUSSION

Our evaluation at Nachusa Grasslands suggests that water table depth is a significant driver of burrowing crayfish populations. However, other predicted variables, such as root biomass and vegetation ground cover, were present in our top models, suggesting some level of influence.

Our findings suggested that canopy cover was not a significant predictor of burrow presence or abundance, however, canopy cover has been significant in previous burrowing crayfish studies (*Rhoden, Taylor & Peterman, 2016*; *Bloomer, Taylor & Distefano, 2022*). This is due primarily to the lack of variation in our largely open-canopy prairie and wetland sampling locations.

Additionally, we standardized root biomass by taking root cores in the top right corner of every quadrat regardless of the plant community or burrow abundance within the quadrat. This method allowed for root biomass comparisons between quadrats. However, this didn't account for the root systems per plant family, plant functional group or root communities surrounding each burrow. This method of standardization may have overlooked the precise relationship between root systems and burrowing crayfish as root systems will vary widely based on family and functional group (*Sperry, 1935*). As such, we recommend taking several root cores and averaging the biomass per quadrat. While root biomass was in our top models it was not a significant variable. The variables selected in this study were determined from previous burrow research and biological intuition, which were designed to focus on the burrowing crayfish populations at Nachusa. For this reason, we suggest future studies modify these variables and methods based on their landscape-specific scale and context.

Our measurement and analysis of quadrat-level habitat variables suggests that fine scale data can provide a better insight into burrowing crayfish habitat associations than large spatial-scale data. Previous studies have looked at the water table depth on a large-spatial scale and found no significant impact on burrowing crayfish populations (*Bloomer, Taylor & Distefano, 2022*; *Quebedeaux et al., 2023*). These studies used the available water storage layer from the gridded Soil Survey Geographic Database (gSSURGO) as a metric for the water table depth. This method provides easily accessible information, but the scale (~30 m) of these data are approximated from digital map data (*Soil Survey Staff, 2022b*; *Rossiter et al., 2022*). While this allows for projections of state- and nation-wide soil data, our results suggest these approximations, in a predictive model context, are too coarse to determine important burrowing crayfish habitat needs. Like most organisms, burrowing crayfish are not uniformly distributed across a landscape, therefore using the wrong spatial scale can overlook significant habitat relationships (*Wiens, 1989*). Future studies may benefit from using water table methods on a landscape-specific scale such as wetland delineation or monitoring wells to target these species habitat associations and ultimately understand burrowing crayfish distributions. Our results further support the importance of boots-on-the-ground to collect fine-scale habitat data to understand a species' habitat needs. Lidar data and other course-scale data should not replace field surveys to create predictive habitat models, instead work in combination with field surveys to generate

accurate predictions (*Rossiter et al., 2022*). Given the predicted impacts on biodiversity from global threats such as climate change and human population expansion, additional research into habitat associations will help to advise and understand the relationship between organisms and their landscapes.

## CONCLUSIONS

This study aimed to determine if habitat variation in a fine-scale context would influence burrowing crayfish populations at a restored prairie site in north-central Illinois. We hypothesized that shallow water table depth, high root biomass, and low canopy cover would predict burrow presence and abundance. Our analysis suggests that at a fine- scale, the Great Plains Mudbug populations in prairie habitats are driven by shallow water table access.

Determining the optimal spatial scale for habitat suitability studies is essential for an accurate investigation into focal species' needs. This is supported by several studies on varying taxa and their respective habitat dynamics (*Wiens, Rotenberry & Van Horne, 1987*; *Kafley et al., 2016*). Primarily because the distributions of a species may be affected by habitat variables at differing resolutions (*Wiens, 1989*; *Price et al., 2005*). For this reason, understanding a species' local abundance and habitat associations can provide valuable insights into their distribution patterns and inform conservation strategies across their native range. Our results become more relevant, as research efforts have promoted prairie and wetland restoration globally which provides a vital refuge for several taxa (*Gleason et al., 2011*; *Bach & Kleiman, 2021*). In the Prairie Pothole Region of the United States alone, over 2 million ha of prairie and wetland habitat have been restored since 1985 (*Gleason et al., 2011*). Such efforts may expand suitable habitats for multiple taxa, but any anthropomorphic changes like construction or alteration to these habitats must be considered at a species-specific scale. For example, modifying a few meters of habitat to create a bike or pedestrian trail, may not appear significant for the greater prairie community. Yet, it could render an entire location unsuitable for an established population of burrowing crayfish if it affects variables such as water table access.

These data were collected within a single nature preserve in Northern Illinois on a single species of burrowing crayfish, the Great Plains Mudbug (*L. nebrascensis*). However, our results may offer valuable information applicable to burrowing species nationwide, as many other burrowing crayfish species tend to demonstrate similar habitat requirements (*Hobbs, 1942*; *Hobbs Jr, 1981*; *Hobbs Jr, 1988*). *Taylor et al. (2019)* suggested gaining a deeper understanding of the fine-scale habitat needs for a single species could then provide valuable insights for multiple, closely related species and those sharing similar habitats. With the challenges in detecting and sampling burrowing crayfish and acquiring funding for non-charismatic species, fine-scale studies, such as this one, could be useful for widespread crayfish conservation efforts. These findings provide support for existing theories on water table depth as a driver for burrowing crayfish, thereby increasing the limited knowledge regarding this elusive taxonomic group. As prairie and wetland conservation initiatives continue to rise globally, it is plausible that burrowing crayfish and many other taxa found

within prairies, will go through significant distributional range shifts. It is imperative that we understand the threshold of their habitat associations to better detect and conserve these species across their native ranges.

## ACKNOWLEDGEMENTS

We thank Caitlin Bloomer for her assistance with field work and model development, Elizabeht Bach for her guidance on experimental design and prairie ecosystem expertise, and Jeffrey Matthews and Cory Suski for their guidance and support.

### Funding

This work was supported by the Friends of Nachusa Grasslands and the Illinois Department of Natural Resources. Funding for article processing charges was provided by the Illinois Department of Transportation and the contents of this document reflect the view of the authors who are responsible for the facts and the accuracy of the data presented herein. The contents do not necessarily reflect the official views or the policies of the Illinois Department of Transportation. The funders had no role in study design, data collection and analysis, decision to publish, or preparation of the manuscript.

### Grant Disclosures

The following grant information was disclosed by the authors:
The Friends of Nachusa Grasslands.
The Illinois Department of Natural Resources.
The Illinois Department of Transportation.

### Competing Interests

The authors declare there are no competing interests.

### Author Contributions

- Molly C. Carlson conceived and designed the experiments, performed the experiments, analyzed the data, prepared figures and/or tables, authored or reviewed drafts of the article, and approved the final draft.
- Dusty A. Swedberg conceived and designed the experiments, performed the experiments, authored or reviewed drafts of the article, and approved the final draft.
- Elizabeth A. Miernicki performed the experiments, authored or reviewed drafts of the article, and approved the final draft.
- Christopher A. Taylor conceived and designed the experiments, authored or reviewed drafts of the article, and approved the final draft.

### Field Study Permissions

The following information was supplied relating to field study approvals (*i.e.*, approving body and any reference numbers):
   Field experiments were approved by the Illinois Chapter of the Nature Conservancy and the Illinois Nature Preserves Commission.

## Data Availability

The raw data are available in the Supplementary File.

## Supplemental Information

Supplemental information for this article can be found online at http://dx.doi.org/10.7717/peerj.17330#supplemental-information.

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
