# Peer review of "Variation in fine-scale water table depth drives abundance of a unique semi-terrestrial crayfish species"

_PeerJ, doi:10.7717/peerj.17330_

## Round 0.1 · original submission · Major Revisions

The reviewers see clear strengths in your manuscript but have also raised a number of concerns that you should address during the revision of your manuscript. Make sure to fully respond to all the major and minor comments and demonstrate how the comments have improved your manuscript.

Reviewer 1 ·

Basic reporting

The manuscript flowed nicely from introduction to conclusion. Literature references with sufficient field background/context were provided.

Experimental design

This paper meets the aims and scopes of PeerJ, focusing on biological science. The research question is well defined, relevant and meaningful and they state how research fills the identified knowledge gap. Enough information was given to be a repeatable study.

This study assesses how fine-scale habitat variables impacts crayfish abundance. This manuscript increases the knowledge of factors that impact primary burrowing crayfish that can be useful in conservation efforts. Because primary burrowing crayfishes are data deficit, this study adds to our knowledge on this group and provides key data on why fine-scale habitat data is essential in their management.

A better explanation of how sites were chosen is needed. One site was sampled without burrows and the text reads as if this site was specifically chosen because there were no burrows present. If this is the case, why was only one site without burrows chosen?

No crayfish were collected in this study. Information needs to be added to address how they knew what crayfish species were in these habitats and if it is truly was only one species.

Date should be added as a factor for statistical analyses. Burrowing crayfishes are often more active during wetter times of the year so sampling into July could impact the number of active burrows at a site. Adding date as a factor will allow us to see if time of year is driving results.

Validity of the findings

All underlying data has been provided; they are robust, statistically sound, & controlled. More discussion should be added about some of the short comings of the study and things that could have impacted the results (more details listed below in general comments).

Additional comments

TITLE
Because this study focuses on crayfish habitat characteristics and abundance, I’d add crayfish to the title. “Variation in fine-scale water table depth drives the abundance of a unique semi-terrestrial crayfish species”

ABSTRACT
Edit sentence starting with “While root biomass and vegetation….”. To “…significant drivers, although they did show in …”
GLMM can be removed because the acronym is not used again within the abstract.

INTRODUCTION
Edit Line 33 to say “The information available demonstrates burrowing crayfish…”
Line 52-55: Sentence is long and convoluted. I would reword to make sentence easier to read.
Reword lines 61-64. Suggested rewording “We aim to evaluate the fine-scale habitat characteristics that may affect the Great Plains Mudbug (Lacunicambarus nebrascensis) (Girard, 1852), a burrowing crayfish species, within a north-central Illinois prairie nature preserve,…”

METHODS
Did you collect any crayfish? How do you know what crayfish species were using these burrows? Are there previous studies that indicate that this is the only species present within Nachusa? Please add information about species identifications or the lack there of.

SITE SELECTION
Line 79: Did you purposefully select one site without burrows or did you select sites and one happened to not have burrows? If the former, why did you select only one site to assess without burrows? If the latter, reword this sentence to state that you selected 6 sites within Nachusa to assess burrowing crayfish populations.

FIELD COLLECTION
Line 96-98: Just to clarify, your transect area was 5 m x 100 m? Were burrow searches just walking and counting burrows along the 5 X 100 m transect? If a burrow was found within 5 m how far away was the next transect. You state a minimum of 10 m, however, I’m not following how the 5 m search comes into play. Please Clarify
Line 100-104: Did you count burrows twice? While doing the burrow searches and in the quadrats? Clarify.

HABITAT VARIABLES
I suggest using water table as the abbreviated version of apparent seasonal high-water table instead of ASHWT because it’s hard to remember what ASHWT refers to.

STATISTICAL ANALYSIS
Crayfish are commonly more active in the spring than summer, which could impact the number of active burrows present in your site. I suggest adding sampling date as a factor in your model.

RESULTS
Line 189-197: You stated in your methods that you documented burrow activity. Please report that information in results.
Line 198: You stated in your methods that your top models have AICc values within 2.5 of the best model. As such Model 1 and 2 are the best binomial models. Model 2 should be included in the text.

DISCUSSION
Line 215: First sentence can be deleted. Just jump into the discussion.
Line 220: Start a new paragraph that discusses root biomass.
Line 222-223: Please add more discussion on how not accounting for root systems per plant family, plant functional group, or root communities around each burrow could affect your results, or how crayfishes may interact with these groups differently.

Information should be added to the discussion that address why you did not physically sample burrows and if this could have impacted your results.

FIGURES
Figure 3: Is the data just displayed in categories (very shallow, shallow, etc.) for this figure, or were categorical differences assessed in the model? Are the depths for these categories based on past studies? If so, add a reference. The categories aren’t mentioned in the text, so some clarification is needed.

Reviewer 2 ·

Basic reporting

TEXT:
I would encourage the author to revise the information provided by previous studies, specifically the Bearden et al 2022 reference. The Bearden et al. 2022 reference did in fact measure fine-scale habitat variables for burrowing crayfish and found that water table depth was a significant driver. That reference is relevant and is not mentioned in that context in this manuscript. The one reference to Bearden et al. 2022 simply notes that monitoring wells were used in the study. Monitoring wells were indeed used to document fine scale variations in burrowing crayfish habitat (water table depth), the same focus and finding as this manuscript. This study differs in that it targets a single species as opposed to multiple species in Bearden et al. 2022.

Also in Lines 35-36, you may also want to note that burrowing crayfish are found in floodplains (Bearden et al. 2022).

FIGURES:
Please include plots for the models with significant variables.

Experimental design

The research question is well defined, and the knowledge gap is identified.
Methods are described with sufficient detail, especially the ASHWT.

Validity of the findings

Impact and novelty assessed but attention should be brought to the fact that water table depth had been previously found to impact burrowing crayfish on a local scale.

Additional comments

Double check manuscript for use of English units, specifically Fahrenheit on line 127.
Revise spelling of Akaike on line 184.

The crayfish community will benefit greatly from this research!

Reviewer 3 ·

Basic reporting

1) No comment
2) Literature references: Overall good, but authors should consider including references where fine scale data has been collected within landscapes at colony sites and/or within transects – though not necessarily at the scale of 1m2 quadrats. See Additional Comments for examples.
3) No comment
4) No comment
5) No comment

Experimental design

1) No comment
2) No comment
3) No comment
4) No comment

Validity of the findings

1) No comment
2) No comment
3) No comment

Additional comments

This is among the best studies that I have seen that examines the relationships between habitat characteristics and burrows. The methodology is extremely thorough and the fine scale of measurements is a real strength. My main concerns revolve around the authors overselling their results in relation to previous studies, but this can be easily fixed.

Introduction
Lines 38 – 50. I have some concerns about this section, in that I think that the impression it gives may be broader than what the authors intended. The authors state that previous studies “..document COARSE-SCALE variables between landscapes, they do not document variation WITHIN a landscape that may drive burrowing crayfish distribution.” This may be true for the previous studies by co-author Taylor that are cited here, but a quick literature search of publications from other burrowing crayfish researchers revealed multiple studies where “within landscape” variables were measured at colony sites and/or along transects and comparisons did not seem limited to only “coarse scale variables between landscapes”. These include (but are likely not limited to) Bearden et al 2022, Freshwater Biology; Helms et al. 2013 Freshwater Science; Loughman et al. 2017, Journal of Crustacean Biology; Loughman et al. 2012, Journal of Crustacean Biology; and Loughman 2010 Southeastern Naturalist. If the authors are trying to say that previous studies have looked at a wide range of within-landscape variables, but few to no studies have looked at variables at the very fine scale of 1m2 quadrats, then I agree. This just needs to be better clarified. Similarly, it is not clear how monitoring wells, which have been used to track water levels monthly at multiple transects within a single catchment (Bearden et al. 2022; Freshwater Biology), showing a strong association between active burrows and depth to groundwater, “…failed to look at fine-scale variations within sites that crayfish inhabit.” Again, this is just a question of clearly defining scale in relation to previous studies. A casual reader may wrongly assume that well(s) were used to examine relationships between water depth and burrows across multiple landscapes (which is not true for Bearden et al. 2022), rather than among transects within a single landscape (which is what it looks to me like what Bearden et al. 2022 actually did and does not seem to qualify as “coarse-scale between landscapes”).
Methods.
Methodology is clearly written, and very well thought out. The ASHWT technique sounds extremely useful. I had not heard of it before and am glad the authors provided enough detail so that others can understand and use it as well. I just had a few suggestions that are easy to address.
Line 112 “dept” should be “depth”
Lines 121 – 124 Suggest providing a reference for canopy cover methodology.
Line 125 Suggest including the appropriate dimension abbreviations. I’m assuming 30 cm is depth and 5 cm is either radius or diameter, but this should be clearly labeled.
Line 131. May not be practical but do the authors think some sort of measurement of root branching/complexity is important? For example, a 100 g mass comprised of a single, cylindrical root may have a much different impact on burrowing than a 100 g mass of finely branching root material.
Line 134. “..using the depth to at least common…” Is this written correctly?
Line 179. “..burrow abundance..” Here and throughout, does “burrow abundance” refer to number of entries or to number of burrows? This is a very important distinction to make since many/most primary burrowers have a variable number of entry holes, and the space between individual burrows is variable. In my experience, it is relatively easy to count number of burrow holes at the surface, but very difficult to determine how many separate burrows are represented by these counts.

Results are clearly described and presented.

Discussion
Line 219. “..due to the lack of variation…” Does this mean that most sites had very little canopy cover?
Lines 228 – 229. Suggest “..association and needs than large spatial-scale studies”.
Lines 228 – 233; 242-242. This is all true, but authors do not mention here that other studies have also made direct measurements (i.e. boots on the ground) near burrow colonies and have also provided some evidence for an association. For example, Bearden et al. 2022 showed a positive correlation between groundwater depth and burrow presence/absence based on groundwater measurements from individual transects on a floodplain. Loughman 2010, Loughman et al. 2017 documented associations between burrows and seeps (i.e. shallow groundwater depths). As stated in comments about a similar issue in the Introduction, I think the authors just need to clarify their main point more clearly. As written, it sounds like this is the first study to find evidence for a relationship between water table depth and burrows using “boots-on-the ground” approaches, but this is not the case. With that being said, I completely agree with the authors that the current study (along with previous studies) illustrate the importance at looking at associations using direct measurements on a fine or very-fine scale as opposed to using Lidar or other coarse-scale data.
Line 237. “course” should be “coarse”
Line 252-253. Would it be better to say something like “..populations of at least some burrowing crayfish species in prairie habitats are driven by…” here since this study focused on a single species and not all burrowing species likely follow the same pattern? Later, the authors suggest that understanding the needs of a single species could provide valuable insights for multiple, closely related species sharing similar habitats. I agree with this statement, but the flip side is that insights from the single species used in this study may or may not apply to less closely related species that inhabit other types of habitats. Thus, Line 252-253 should probably be modified.
Lines 278-279. “These findings substantiate existing theories that have lacked support…” Authors should clarify which existing theories they are referring to here. The theory that burrowing crayfish prefer habitats exhibiting shallow groundwater depths already had support from previous studies mentioned in this review that directly measured groundwater depth at burrow colony locations using wells and/or the presence of seeps. I completely agree that this study supplies valuable additional support for this theory, but I’m not convinced the groundwater depth theory previously lacked support. The theory that burrows are related to canopy cover was not substantiated by this study – the finding was that burrows were not related to canopy cover. The theories that burrows are related to vegetation cover and/or root biomass received only weak support from this study – while these variables were present in top models they were not significant predictor variables. Thus it may be a stretch to say that these findings SUBSTANTIATE the vegetation and root biomass theories that previously lacked support.
This is just my long-winded way of saying authors may want to reword this section as something like “These findings provide further evidence for…shallow groundwater depth….do not support (relationship with canopy cover)…and provide some initial evidence for (root mass, vegetation cover)..”

---

## Round 0.2 · Minor Revisions

You need to address the additional comments by the reviewers before I can make a final decision on your manuscript.

Reviewer 1 ·

Basic reporting

The manuscript flowed nicely from introduction to conclusion. Literature references with sufficient field background/context were provided. All comments and suggested edits have been addressed which has clarified all questions addressed in my review.

Experimental design

This paper meets the aims and scopes of PeerJ, focusing on biological science. The research question is well defined, relevant and meaningful and they state how research fills the identified knowledge gap. Enough information was given to be a repeatable study.

This study assesses how fine-scale habitat variables impacts crayfish abundance. This manuscript increases the knowledge of factors that impact primary burrowing crayfish that can be useful in conservation efforts. Because primary burrowing crayfishes are data deficit, this study as to our knowledge on this group and provides key data on why fine-scale habitat data is essential in their management.

Validity of the findings

All underlying data has been provided; they are robust, statistically sound, & controlled.

Additional comments

Line 117 --I believe a typo has been made in regards to the size of the larger quadrat. It is listed as "...a larger 1 m2 quadrat..." However, I think it should state 9 m2. Please revise accordingly.

Sentences shouldn't start with acronyms. Please change throughout the paper.

Line 204--While c-hat does check for over dispersion marginal and conditional r2 does not. Please reword the sentence to clarify what marginal r2 and conditional r2 measure.

Line 205-206 Sentence should be reworded to show that you used AICc and not AIC. "Candidate models were created and evaluated using the standard correction to Akaike's information criterion (AICc)..."

Reviewer 3 ·

Basic reporting

This is a second review of a revised manuscript. The authors have done a very nice job addressing my previous concerns and addressed them to my satisfaction. I have no further concerns/edits

Experimental design

This is a second review of a revised manuscript. The authors have done a very nice job addressing my previous concerns and addressed them to my satisfaction. I have no further concerns/edits

Validity of the findings

This is a second review of a revised manuscript. The authors have done a very nice job addressing my previous concerns and addressed them to my satisfaction. I have no further concerns/edits

Additional comments

This is a second review of a revised manuscript. The authors have done a very nice job addressing my previous concerns and addressed them to my satisfaction. I have no further concerns/edits

---

## Round 0.3 · accepted · Accept

The reviewers have implemented all the comments raised by the reviewers and I judge the manuscript acceptable for publication.